# Fatigue Performance of Thin Laser Butt Welds in HSLA Steel

Patricio G. Riofrío [1], Fernando Antunes [2,*], José Ferreira [2], António Castanhola Batista [3] and Carlos Capela [2,4]

1   DECEM, Departamento de Ciencias de la Energía y Mecánica, Universidad de las Fuerzas Armadas-ESPE, Av. General Rumiñahui S/N, Sangolquí 171103, Ecuador; pgriofrio@espe.edu.ec
2   CEMMPRE, Department of Mechanical Engineering, University of Coimbra, 3030-788 Coimbra, Portugal; martins.ferreira@dem.uc.pt (J.F.); carlos.capela@ipleiria.pt (C.C.)
3   CFisUC, Department of Physics, University of Coimbra, 3004-516 Coimbra, Portugal; castanhola@uc.pt
4   ESTG, Department of Mechanical Engineering, Instituto Politécnico de Leiria, Morro do Lena—Alto Vieiro, 2400-901 Leiria, Portugal
*   Correspondence: Fernando.ventura@dem.uc.pt; Tel.: +351-9347-98913

**Abstract:** This work is focused on understanding the significant factors affecting the fatigue strength of laser-welded butt joints in thin high-strength low-alloy (HSLA) steel. The effects of the weld profile, imperfections, hardness, and residual stresses were considered to explain the results found in the S-N curves of four welded series. The results showed acceptable fatigue strength although the welded series presented multiple-imperfections. The analysis of fatigue behavior at low stress levels through the stress-concentrating effect explained the influence of each factor on the S-N curves of the welded series. The fatigue limits of the welded series predicted through the stress-concentrating effect and by the relationship proposed by Murakami showed good agreement with the experimental results.

**Keywords:** fatigue strength; laser butt weld; HSLA steel; fatigue limit prediction





## 1. Introduction

High-strength steels have high fatigue resistance to crack initiation, as this damage mechanism decreases with the increase of the material's tensile strength. The welding of these materials may be done using laser welding due to low heat inputs, small heat-affected zones, and less softening of the material. However, reductions of up to 40% in the fatigue limit, low ratios of the fatigue limit to the ultimate strength of the base metal (BM), and fatigue limits that do not exceed 175 MPa were reported in [1–3] when thin high strength steels were laser welded.

The relevant facts that comprise the differences between welded joints and elements with notches or other defects, are the presence of crack-like imperfections and the changes introduced by the heat input (HI) into the BM. This has led to the fatigue initiation period being neglected [4,5] and to the application of the fracture mechanics approach, considering only the period of crack growth for the assessment of the fatigue life [6]. However, for small sized cracks and for low stress levels, the initiation stage is considered [7,8].

The conventional approach used to predict the fatigue limit in notched elements based on stress concentration and stress gradient models is also used in welded joints [9]. In [10,11], several traditional models of sensitivity to notches and new models were analyzed for the prediction of the fatigue limit of elements with small imperfections and for the application in welded joints. Based on fracture mechanics, Murakami [12] proposed an expression for fatigue limit predictions of materials containing small defects or cracks as a function of the hardness and the size of defects, and Åman et al. [13] modified the mentioned expression to consider the effect of the notch root on the small defects and suggested the application to the welded joints.

Several factors affecting fatigue strength in weldments are highlighted in [4,6,14], however, they can be summarized in the following: weld quality, residual stresses, material,

and size effects. Although it has been shown that the weld quality control [15] or the use of post-weld treatments [16–19] in welded joints with high-strength steels increases the fatigue strength by reducing the severity of its inherent imperfections, this is recognized in design guides, allowing the increase of fatigue class curve (FAT) when the imperfections were verified with nondestructive testing (NDT) or when an improvement method was used [5]; there is no differentiation in the increase according to the mechanical strength of the materials. The foregoing has been explained due to the fact that the growth of cracks dominates the fatigue life of the welded joints and because the fatigue crack growth rate (FCGR) is insensitive to the mechanical strength of the material [20]. However, as counterpart, it was shown that when the weld quality is high, for high-strength steels, fatigue strength is positively correlated to mechanical strength [21].

Microstructural features such as phases and hardness are factors influencing the fatigue behavior of the welded joints. Due to the welding process, the microstructures of the heat-affect zone (HAZ), fusion zone (FZ), and BM are different and since, in most cases, the fatigue starts and paths are in the HAZ or FZ, the properties of these zones are important. Thus, the hardness is used instead of those of the BM, improving the results or fatigue predictions as shown by Kucharczyk et al. [22]. In [23], it is shown that the increase of the proportion of martensite in the microstructure decreases the FCGR and increases the crack growth threshold. In welded joints, the size effects refer to the decrease in fatigue strength when the thickness of the plates increases [4]. For this reason, fatigue strength reduction factors are applied when 25 mm of thickness is exceeded [5]. Although there is evidence that the opposite effect occurs for thin thicknesses [7], this is not considered in the design guides.

The general effect of notches and residual stresses on fatigue strength of welded joints is well known. In various studies, their combined effects were studied and a relative effect was observed. Watanabe et al. [24] reported that the fatigue strength is dominated by stress concentration factor (SCF) when this is high and by the residual stresses when SCF is small. Harati et al. [25] concluded that the residual stresses have a larger influence than the weld toe geometry, meanwhile, Nguyen et al. [26] established that for tensile residual stresses of low magnitude or for compressive stresses, the effect of an undercut is dominant. Shiozaki et al. [27] reported that due to the equality of results in the S-N curves between the as-welded and heat-treated specimens, the effect of residual stress should not be considered. In almost all previous works, relatively thick plates and conventional welding processes were used.

There are few studies on the main factors influencing fatigue strength and on the application of the fracture mechanics approach to the evaluation of fatigue limits of thin steel plates in laser welded joints. In the work [28], for butt joints of 3 mm thickness in Strenx® 700MCE steel welded by laser, the weld bead geometry, the imperfections, and the quality level of the welds according to the ISO 13919-1 [29] standard were determined. The fatigue strength of the mentioned welded joints was evaluated in this study. For this, local properties such as hardness and residual stresses in the HAZ and FZ and a more realistic weld profile will be considered. The actual profile of the weld beads, including imperfections, is completely modeled to determine the SCFs by FEM. The residual stresses are measured by X-ray technique. Through the failure analysis of fractured surfaces and using the stress-concentrating effect, the influence of the main factors on fatigue behavior at low stress levels is explained. Predictions of the fatigue limits of the welded series are also rendered both by the stress-concentrating effect and by the relationship proposed by Murakami.

## 2. Material and Procedures

### 2.1. Material and Laser Welding

Butt weld joints were created with 3 mm thick high-strength low-alloy (HSLA) steel plate Strenx® 700MCE [30]. This steel has the chemical composition and tensile mechanical properties shown in Tables 1 and 2 [31], respectively. It presents a fine-grain ferritic-bainitic

microstructure. The yield stress is close to the tensile strength, which indicates that this material keeps the elastic behavior almost until final fracture.

**Table 1.** Base metal chemical composition (wt %).

| C | Mn | Si | P | S | Cr | V | Nb | Ni | Cu | Al | Mo | Ti | Co | Fe |
|---|---|---|---|---|---|---|---|---|---|---|---|---|---|---|
| 0.07 | 1.69 | 0.01 | 0.012 | 0.006 | 0.03 | 0.02 | 0.046 | 0.04 | 0.011 | 0.044 | 0.016 | 0.117 | 0.016 | balance |

**Table 2.** Mechanical properties of the base metal.

| Yield Strength (MPa) | Tensile Strength (MPa) | Elongation (%) |
|---|---|---|
| 808 | 838 | 15.0 |

Five series were welded considering the welding parameters shown in Table 3. The S1, S2, S3, and S4 series are single-welded joints while the S5 series is double-welded joint. For this last series, the welding speed of the weld pass on the top side was different for each sample (1.75, 1.80, 1.90, 1.95, and 2.00 m/min), while the weld pass on the bottom side was constant for all samples (2.50 m/min). These parameters were selected based on a previous study of the effect of parameters on defects and tensile properties [31]. The heat inputs used in welding were defined in order to achieve different values of hardness in the welded series, but simultaneously maintaining small HAZs, low softening, and tensile mechanical properties similar to the BM, as shown in [31]. A disk laser equipment Trumpf TruDisk 2000 with: laser maximum output, 2000 W; beam wavelength, 1020 nm; beam parameter product, 2 mm-mrad; and fiber diameter, 200 µm was used in continuous mode. Argon (99.996%) shielding gas was supplied only on the top side of the weld at a flow rate of 20 L/min. Figure 1a shows the size of plates used in the welding process.

**Table 3.** Welding parameters used in the experimental work.

| Series | Laser Power (kW) | Welding Speed (m/min) | Heat Input (J/mm) |
|---|---|---|---|
| S1 | 2.00 | 1.60 | 75.0 |
| S2 | 1.75 | 1.60 | 65.6 |
| S3 | 2.00 | 2.00 | 60.0 |
| S4 | 1.75 | 2.00 | 52.5 |
| S5 | 1.75 | top side weld pass 1.75–2.00 | 52.5–60.0 |
| | 1.25 | bottom side weld pass 2.50 | 30.0 |

In work [28], according to the welding quality standard ISO 13919-1, a B quality level for the S1, S2, and S3 welded series and a D quality level for the S5 welded series, were reported and according to the imperfection sizes reported, these can be considered as crack-like imperfections (see the imperfections sizes in Table 6). In the aforementioned work, the imperfections were measured using a profilometer with a pitch of 1 µm and a microscope equipped with digital micro-meters. Since the S4 series did not reach full penetration, it will be omitted in the next sections. Therefore, this series defines approximately the lower boundary of the heat inputs in order to obtain full penetration of the welding in the 3 mm thick specimens.

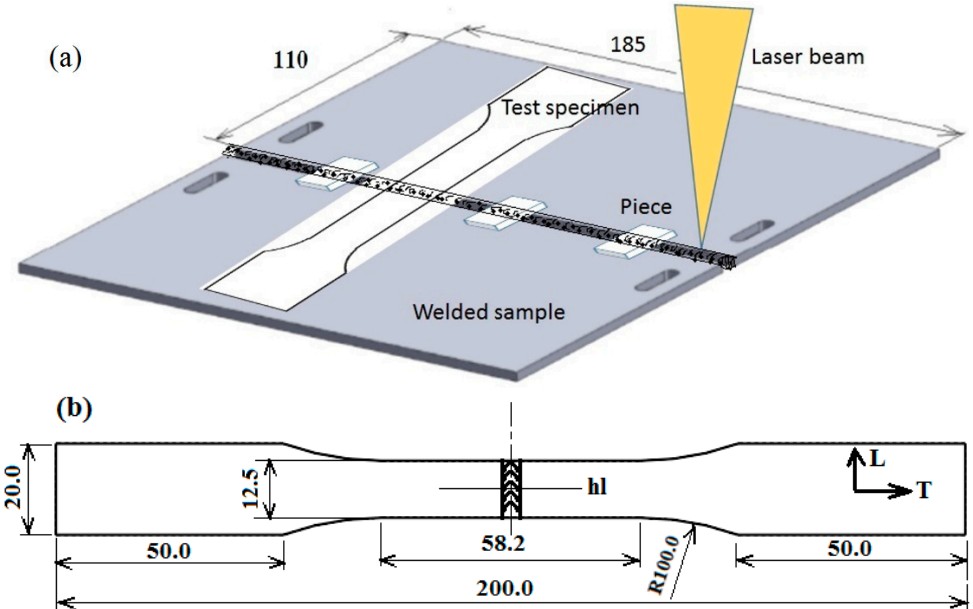

**Figure 1.** (**a**) Sketch of plates (dimensions in mm) for the welding and location of specimens, (**b**) dimensions (in mm) of specimens used in fatigue tests.

### 2.2. Fatigue Testing and Analysis

The fatigue specimens were prepared and tested according to the guidelines of the ASTM E466-15 standard [32]. Figure 1a illustrates the location of the specimens in the welded plates, which were obtained by laser cut, meanwhile, Figure 1b shows specimens' dimensions. The fatigue specimens were tested in three conditions: not-welded (BM), as-welded, and bottom side-removed. For this last condition, the excess weld metal and undercuts were progressively eliminated in order to obtain a smooth transition between the weld and the plate, thus minimizing the stress concentration. The specimens were carefully ground to a final polish with sandpaper # 220, particularly the edges and the welding surfaces. The total elimination of the undercuts was verified using the optical microscope with 20× magnification power.

The testing was performed using DARTEC INSTRON servo-hydraulic machine. The fatigue specimens were axially loaded with constant amplitude at a stress ratio R = 0 and at a frequency of 15 Hz. The specimens were tested up to the final fracture at various stress levels; however, when at a certain stress level in which the life of the specimens exceeded $2 \times 10^6$ cycles, the tests were stopped (run-outs). After the fracture of the specimens, their surfaces were analyzed with a stereo-microscope in order to determine the site, start, imperfection, path, and shape of the fatigue surface.

A power law relationship was assumed for the S-N curves:

$$\log (N) = \log (C) - m \log (\Delta\sigma), \tag{1}$$

where the nominal stress range $\Delta\sigma$ is the independent variable, the number of cycles N at which the failure occurs is the dependent variable and C, m are parameters that characterize the curves. Through linear regression, the S-N data were analyzed according to the ASTM E739-10 standard [33] to determine the parameters (C, m) of the S-N relationships, the median S-N curve, and its 95% confidence band. The testing of adequacy of the linear model was also verified according to the mentioned standard. For comparison, the FAT100 curve (slope m = 3, $\Delta\sigma$ = 100 MPa at two million cycles) was also plotted in the S-N graphs. This FAT100 curve, suggested in the fatigue design recommendations of the International Institute of Welding (IIW) for butt joints when imperfections (undercuts and porosity) are evaluated [5], was chosen because its requirements closely match (differs in the thickness

and the porosity of one of the series) the imperfection sizes and quality level found in welded series, as reported in [28].

### 2.3. Measurement of Residual Stresses

The residual stress analysis was performed by X-ray diffraction using Proto iXRD equipment in the longitudinal and transverse directions of the laser welds and on both sides (top and bottom surfaces) of the specimens. Lattice deformations of the {211} diffraction planes ($2\theta \approx 156°$) were measured using Cr-K$\alpha$ X-ray radiation, with 11 $\beta$ angles in the range $\pm30°$ (22 $\psi$ angles in the range $\pm42°$), an acquisition time of 30 s by peak, and $\pm2°$ oscillations in $\psi$. An aperture length of 0.5 mm $\times$ 3 mm in the transverse and longitudinal directions, respectively, was used. The residual stresses were evaluated with an elliptical regression of $\sin^2\psi$ using the X-ray elastic constants of $5.83 \times 10^{-6}$ (MPa)$^{-1}$ for $\frac{1}{2}S_2^{\{211\}}$ and $-1.28 \times 10^{-6}$ (MPa) $^{-1}$ for $S_1^{\{211\}}$. For the analyzed material and considering the radiation used, the average penetration depth of the X-rays was about 5 μm.

The measurement of the residual stresses was performed in the middle of the width of the fatigue specimen at points located along the horizontal line (hl), see Figure 1b. A 32 mm extension was covered with a separation between points that was progressively reduced when approaching the welded zone, these separations were: 4 mm, 2 mm, 1 mm, and 0.5 mm. Figure 1b shows the directions of the longitudinal (L) and transverse (T) residual stresses.

### 3. Results and Discussion

### 3.1. S-N Curves

Figure 2 shows the S-N experimental results of both the welded series and the BM. The mean and the 95% confidence band curves of the BM are also presented along with FAT100 curve, which is a reference for welded joints. The BM shows a linear behavior in log-log scale with small scatter. The nominal stress range varied between 600 and 800 MPa, which corresponds to 74% and 99% of the material's yield stress, respectively. The modifications introduced by laser welding on microstructure significantly changed the position of the S-N curves. A significant decrease of stress range for the same fatigue life is evident, namely for lower load ranges, i.e., for longer fatigue lives. The scatter increased significantly and comparatively to the BM. Both the S-N curves of the BM and of the welded joints presented the characteristic "knee point" that identifies the stress level of the fatigue limit, which, in this work, will be assumed as the fatigue strength of run-outs at two million cycles. Eventually, the series S2 may be considered to have a higher limit. The FAT100 curve is clearly below the experimental results obtained for the welded series, which is a good indication for the quality of laser welding.

Figure 3 presents the results for each series and a photo-macrograph showing the typical cross-section of the weld bead. The stress levels corresponding to the fatigue limit are also indicated. The dispersion of the entire data set is reduced when each series is considered separately, and the dispersion of each series is similar to that of the BM. In the case of the S5 series, because two specimens (named as S5-F9) showed considerably higher fatigue strength than the rest of specimens, they were not considered within the S5 series. The FAT100 is clearly below the experimental results of all the series.

Table 4 presents log C and m parameters of S-N curves. There is a limited influence of welded series on these values, with the exception of S2 series. The m values are close to FAT100 curve slope. The relatively high m value obtained for the BM indicates that the initiation is dominant over the propagation. Conversely, the relatively low value of m for the welded joints indicates the predominance of propagation over initiation. The fatigue limits shown in Table 4, which were assumed to be the maximum stress range values reached by the specimens that exceeded two million cycles, are between 30% and 57% of the fatigue limit of the BM. As can be seen in Figure 3, all series had a fatigue life over the FAT100 curve. From the results obtained, the confidence F-test for all welded series and BM did not reject the hypothesis of linear model.

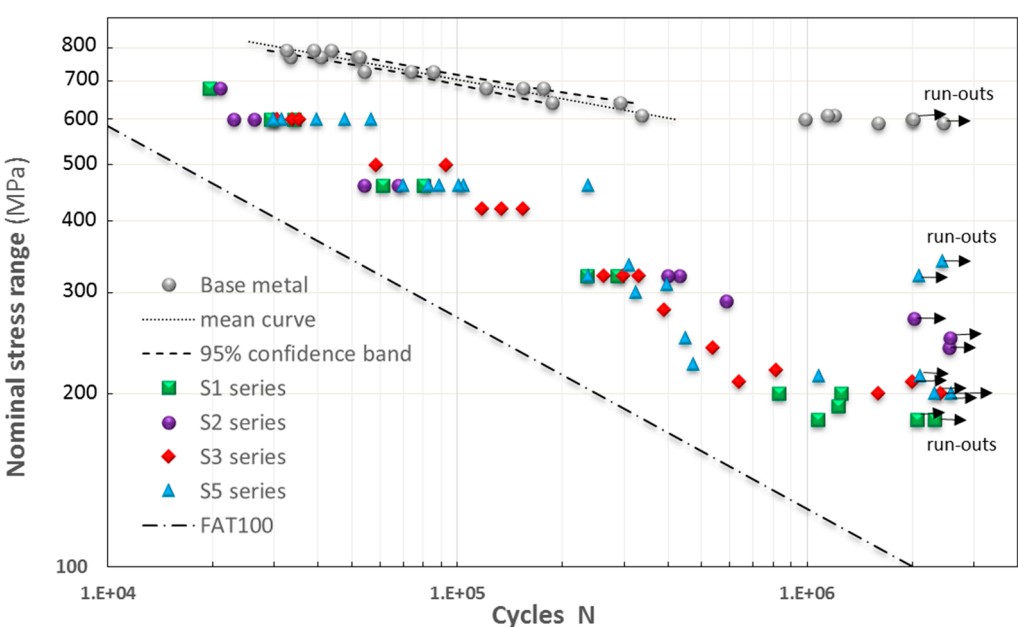

**Figure 2.** S-N curves for all welded series and BM.

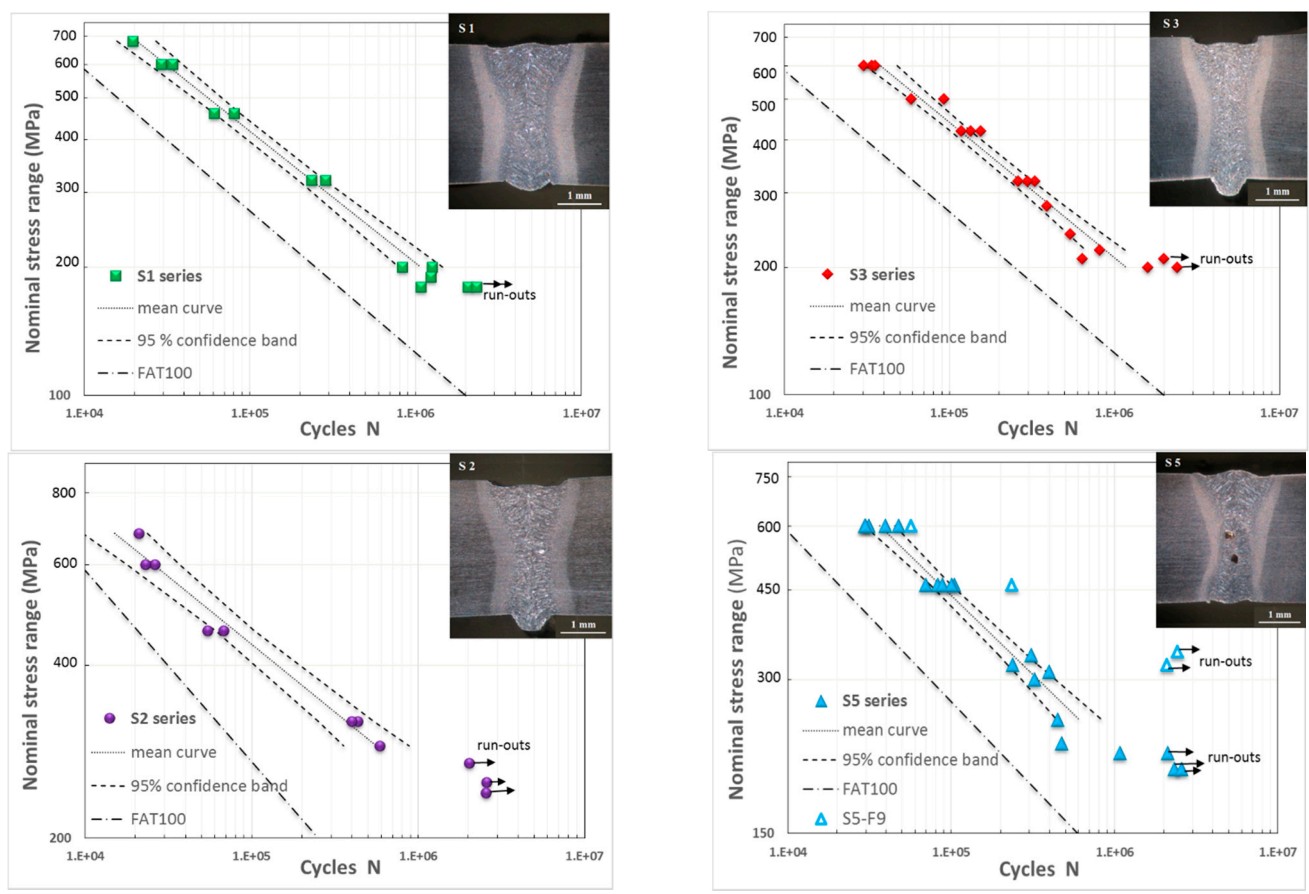

**Figure 3.** S-N curves of each welded series.

**Table 4.** S-N curves parameters and fatigue limits.

| Series | Linear Model F Test $F_{0.95}$ $F_{calculated}$ | log C | m | Fatigue Limits (MPa) |
|--------|---------------------|-------|------|---------------------|
| S1 | 6.59 0.27 (not rejected) | 13.46 | 3.23 | 180 |
| S2 | 9.28 7.86 (not rejected) | 16.21 | 4.25 | 270 |
| S3 | 3.97 1.62 (not rejected) | 13.25 | 3.12 | 210 |
| S5 | 3.69 1.57 (not rejected) | 13.47 | 3.21 | 215 (340) [1] |
| BM | 3.70 0.44 (not rejected) | 30.64 | 9.00 | 600 |

[1] The value 340 MPa corresponds to S5-F9 specimens.

### 3.2. Fatigue Failure Modes

Based on the analysis of the fractured surfaces of all the specimens, mainly four failure modes were found in the welded series. These are designated as FM1, FM2, FM3, and FM4. Table 5 presents the principal features of each failure mode: the series in which the failure mode was present; the site, imperfection, and side of the weld bead where fatigue started; details of the imperfections; the shape of the fractured surface; and remarks. Table 5 includes representative photographs showing the fracture surfaces, the fatigue initiation sites, and details of the imperfections that caused the fatigue. For each mode, the photographs of two fractured specimens are presented with the exception of the FM4 mode, for which only the photographs are presented for one specimen.

**Table 5.** Failure modes features.

| Failure Mode (Series) | Fatigue Start Site-Imperfection (Side) | Details | Approximate Shape of Fractured Surface | Remarks |
|-----------------------|----------------------------------------|---------|----------------------------------------|---------|
| **FM1** (S1, S3) | root-undercuts (bottom side) | (a) long and deep undercuts, one or two very close (b) several undercuts across the width | (a) big semi-elliptical (b) tending to rectangular by coalescence of small semi-elliptical | in some cases, microdefects close to undercuts |

**Table 5.** *Cont.*

| Failure Mode (Series) | Fatigue Start Site-Imperfection (Side) | Details | Approximate Shape of Fractured Surface | Remarks |
|---|---|---|---|---|
| **FM2** (S2, S1, S3) | root-excess weld (bottom side) | multiple starts in micro-imperfections (pores, cavities, shallow crack-like) across the width | tending to a rectangular stripe across the width | semi-elliptical forms were not observed in weld root |

| FM3 (S5) (S1, S2, S3) | face-underfill (bottom side) (top side) | multiple starts in waves and sharp ripples with microdefects | small semi-ellipses or tending to a big semi-ellipse by coalescence | in many cases, underfill at border |
|---|---|---|---|---|

**Table 5.** *Cont.*

| Failure Mode (Series) | Fatigue Start Site-Imperfection (Side) | Details | Approximate Shape of Fractured Surface | Remarks |
|---|---|---|---|---|
| **FM4** (S5, S1, S2, S3) | center-porosity (lateral side) | single pore or several on or near the surface | approaching the middle of a semi-elliptical | in some cases influenced by underfill close to corner |

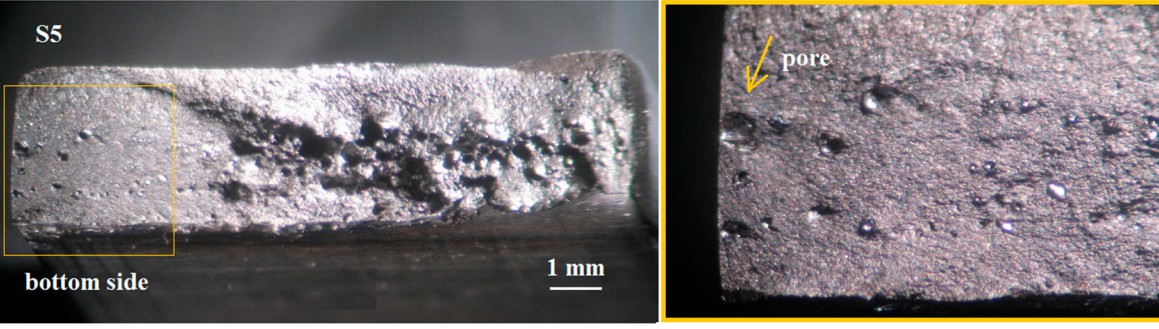

The FM1 failure mode was due to the undercuts at weld root of the S1 and S3 series; there was generally more than one undercut that generated semi-elliptical cracks. In the FM2 failure mode, multiple imperfections along the weld root caused failures, especially in the S2 series. In these cases, the presence of semi-elliptical cracks was not evident. The FM3 failure mode was present in practically all the welded series and the fatigue starts were located in the underfill and the ripples. The ripples corresponding to the S5 series were smaller; see Table 5. The FM4 failure mode refers to failure due to the presence of pores that were present in all series but in greater numbers in the S5 series. The pores caused fatigue failures at the lateral side of the specimens generally when they were located near or on the surface. This failure can also be influenced by the underfill at the border of the element, as illustrated in Table 5.

It should be noted that in approximately 44% of the fractured specimens, at least two failure modes were observed (as illustrated in FM2 mode, S3 series in Table 5), however, in most cases, only one mode was decisive to fatigue failure. The fatigue path was generally straight from the weld root through thickness to the underfill or vice versa. However, in certain cases, the path had inclined parts due to the starts of fatigue from different planes by the presence of porosity, or in the final stages of the fracture, due to deviations to the soft material where greater deformations were presented.

Figure 4 presents the failure modes observed in each specimen of the different series. Each series had a main failure mode that prevailed over the others, and all series presented at least two main and two additional failure modes. As can be seen, in S1 and S3 series the FM1 mode prevailed, in S2 series the FM2 mode is more relevant, while in S5 series the FM3 mode is dominant. The above is strongly related to the corresponding presence of imperfections in each weld bead; see Table 5. Figure 4 also illustrates, in the sketches on the cross-sections of the weld beads, the starts and paths followed by the four failure modes and that all the fractures in the specimens occurred within the FZ and HAZ. Considering the starts on underfill and the starts on excess weld and undercuts, it was observed that the last dominated in S1, S2, and S3 series while the first dominated in S5 series. The FM4 mode, which corresponds to porosity, caused few starts in all series, however, in S5 series it increased as a consequence of its highest percentage of porosity (3.5%) [28].

### 3.3. Effect of Weld Bead Geometry and Imperfections

In order to evaluate the effect exerted by the weld profile and geometrical imperfections on the fatigue behavior, the stress concentration factor (SCF), $K_t$, corresponding to the top and bottom sides of each welded series were determined by the finite element method

(FEM) and with expressions from the literature. For the latter case, SCF values shown in Table 6 were calculated with semi-analytical expressions proposed in [34] for the weld profile, notches, and their interaction, considering the mean values of the geometric parameters presented in Table 6. The ANSYS commercial code was used for the FEM. Slice models (0.2 mm in thickness) with meshes composed of quadrilateral elements with quadratic displacement function were generated according to the recommendations regarding the number of elements in notches and weld toes given in [35], i.e., element size $\leq 0.012$ mm (for radius 0.05 mm); number of elements over $45°$ arc $\geq 3$ and for $360°$ arc $\geq 24$. Previous to the FEM, the weld profiles (top and bottom sides) of the four welded series were completely modeled according to the procedure and the mean geometrical parameters given in Table 6 and reference [28]. The models are illustrated in Figure 5, which also show the fine mesh used in the models and in the imperfections of the weld root (toe), undercut, and underfill. Therefore, the number of elements was exceeded, and the sizes (approximately 0.010–0.015 mm) were close to the recommendations mentioned. To calculate $K_t$, a stress of 0.3 MPa was applied to the lateral edge of the slice model while the opposite edge was defined as a fixed support. The SCFs calculated by FEM according to the indicated procedure are shown in Table 6. The SCFs were established as already mentioned through the mean values of the imperfections, and the effect of their variation on the $K_t$ values was not determined.

Figure 6a,b illustrate the maximum principal stresses for the bottom side of the S3 and S5 welded series obtained by means of FEM, according to the procedure detailed above. As shown in Figure 6, the stress distribution of the imperfections are different, being more concentrated with a major gradient for the undercut, while the underfill has a wider and smaller gradient, as expected due to the difference in notch radii. In general, there is a difference of less than 10% between the SCFs determined with the semi-analytical expressions and by FEM. Conversely, the SCFs corresponding to the undercuts are almost double the SCFs for the underfill or excess weld (toe), and there is some closeness between all the SCFs due to underfill and excess weld with exception of the value of the S5 ($K_t = 1.36$).

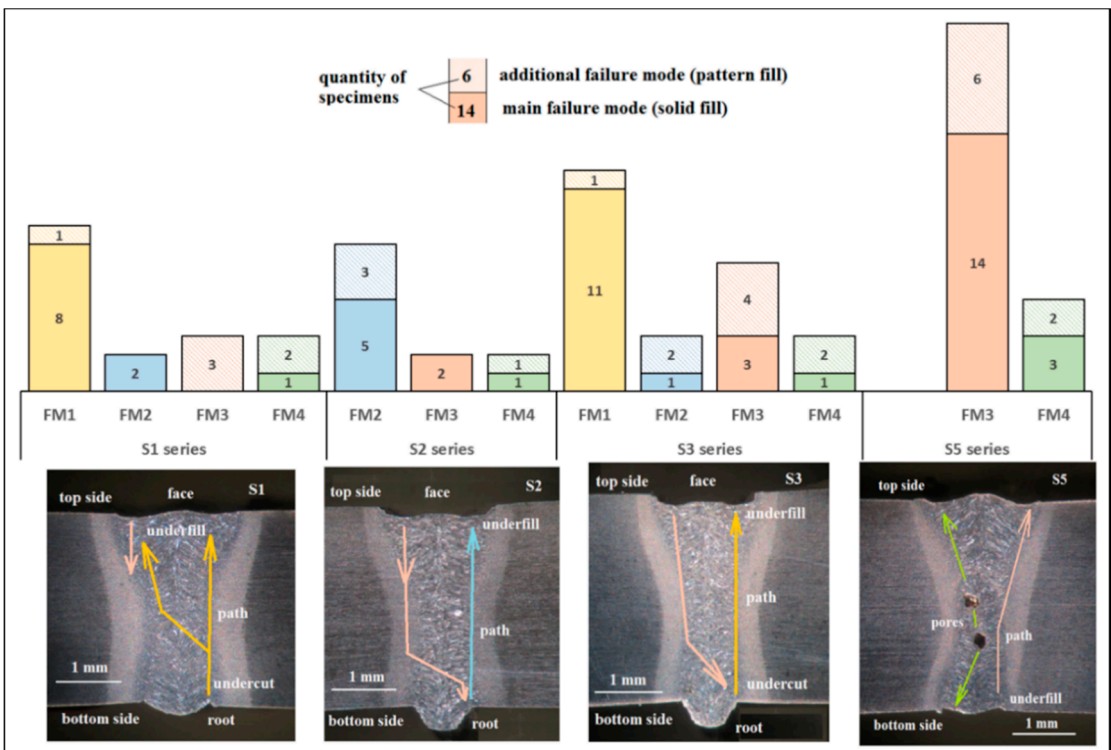

**Figure 4.** Failure modes distributions found in the specimens of the welded series and sketches of starts and paths on their cross-sections.

**Table 6.** SCFs and mean geometric parameters of imperfections.

| | | | Welded Series | | | |
|---|---|---|---|---|---|---|
| | | | **S1** | **S2** | **S3** | **S5** |
| Excess weld | radius | (mm) | 0.16 | 0.06 | 0.07 | 0.54 |
| (toe) | width | (mm) | 1.02 | 0.93 | 0.84 | 1.73 |
| | high | (mm) | 0.20 | 0.36 | 0.31 | 0.14 |
| | flank angle | (degrees) | 34 | 50 | 50 | 14 |
| | SCF analytical | | 1.54 | 1.95 | 1.83 | 1.30 |
| | SCF by FEM | | 1.71 | 2.25 | 1.98 | 1.36 |
| Undercut | depth | (mm) | 0.04 | - | 0.05 | - |
| | radius | (mm) | 0.06 | - | 0.03 | - |
| | SCF analytical | | 3.56 | - | 4.89 | - |
| | SCF by FEM | | 3.29 | - | 4.41 | - |
| Underfill | depth | (mm) | 0.08 | 0.12 | 0.15 | 0.06 |
| | radius | (mm) | 0.53 | 0.60 | 0.50 | 0.27 |
| | SCF analytical | | 1.72 | 1.84 | 2.05 | 1.91 |
| | SCF by FEM | | 1.75 | 1.77 | 1.94 | 1.95 |

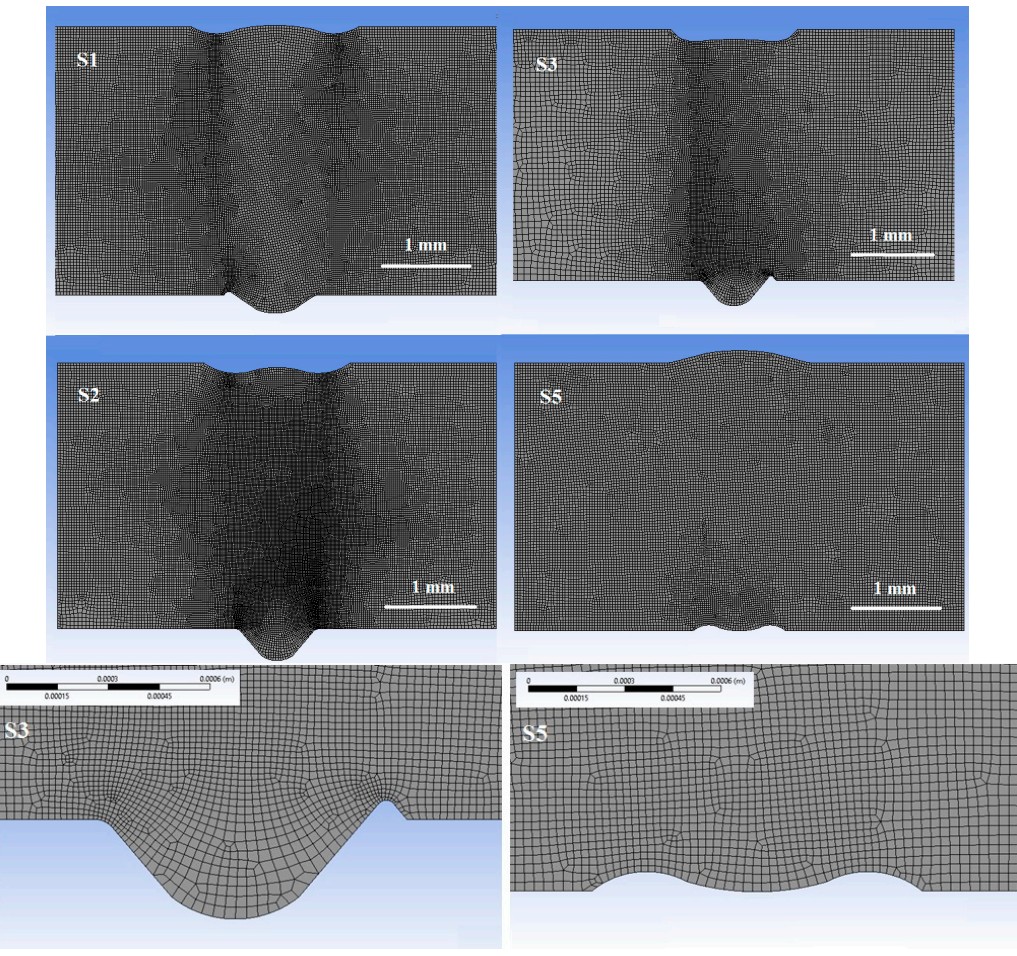

**Figure 5.** Weld profile models of the welded series and details of meshing used in FEM.

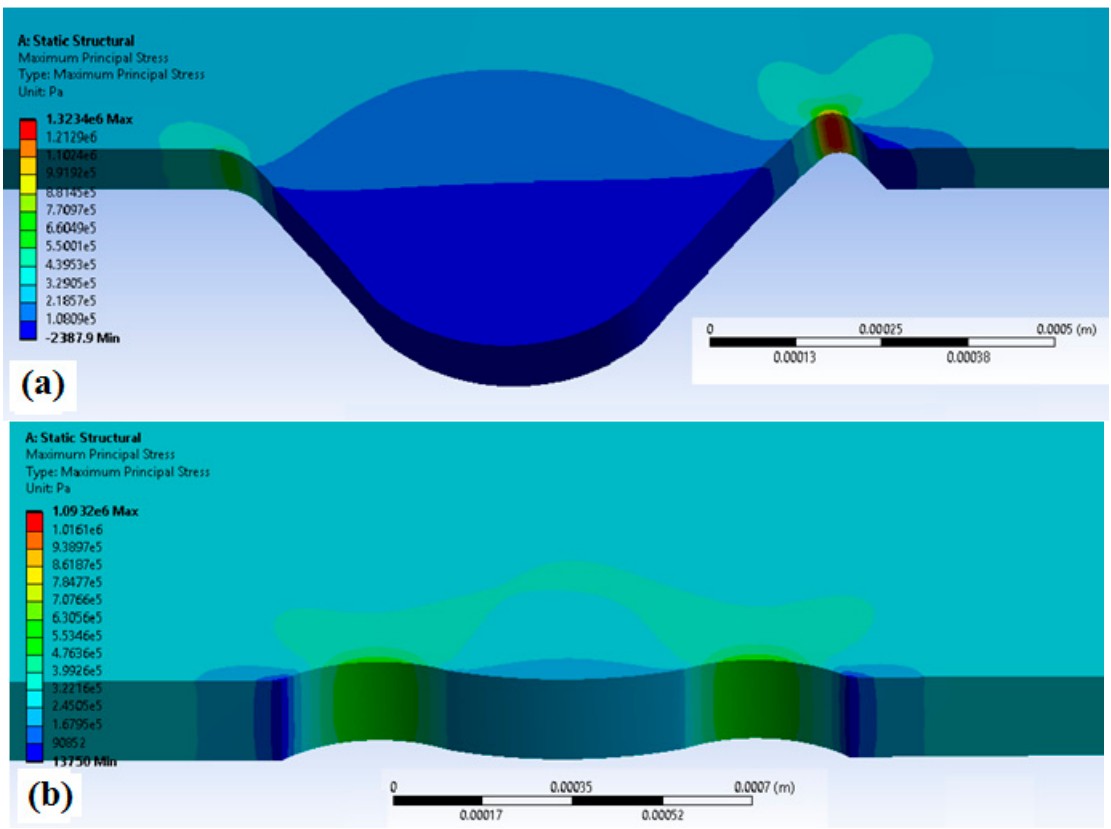

**Figure 6.** Stress distribution in bottom side of the welded series: (**a**) toe and undercut, S3 series and (**b**) underfill, S5 series.

When considering the SCFs of the imperfections in Table 6 and the failure modes in Table 5, the highest SCFs in each welded series are correlated to their major main failure mode and the respective imperfection, thus: 3.27-undercut-S1, 2.25-excess weld-S2, 4.41-undercut-S3, and 1.95-underfill-S5. The foregoing highlights the known fact that fatigue cracks commonly start in sites of stress raisers.

### 3.4. Effect of Top Side and Bottom Side on the Fatigue Strength

Because the weld beads of the welded series showed fatigue starts on both sides, to better examine the effect of weld profile and imperfections on fatigue behavior, various specimens were tested once the bottom side was removed from their weld beads. Figure 7 shows the S-N results for specimens of the S1 and S2 series in two conditions: as-welded and bottom side-removed. In S1 series (Figure 7a) the elimination of the bottom side significantly increases the fatigue strength for all stress ranges. Therefore, the slope of S-N curve is not significantly affected. As such, the fatigue limit increased from 180 to 380 MPa.

In S2 series (Figure 7b) there is a significant effect of $\Delta\sigma$. At high stress range, the change of geometry does not significantly affect the fatigue life. Conversely, at low stress range there is a significant effect on fatigue life. Therefore, there is a great reduction in the slope of S-N curve, which indicates a change of dominance from propagation to initiation. The smoothing of geometry also increases the fatigue limit from 270 to 390 MPa.

These results can be expected since the main imperfections (excess weld and undercuts) were eliminated, which significantly reduces the SCF. Therefore, the fatigue initiation changed to the top sides, where the stress raisers are less severe. Table 7 presents the fatigue limits and the corresponding SCFs for series S1 and S2, as-welded and with bottom-side removed. The fatigue strength at two million cycles was substantially improved by eliminating defects from the bottom side of the weld beads. In average, the increase of two welded series reached 64% of the fatigue limit of the BM. However, at high stress levels the

increase of the fatigue life is smaller for the S2 series. A well-defined trend exists between the SCF and fatigue limit, which reinforces the importance of this parameter.

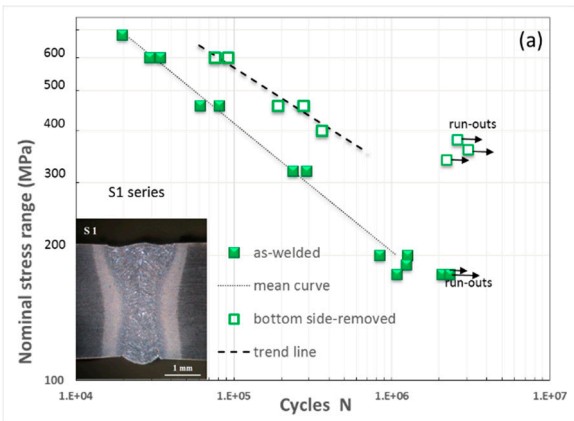 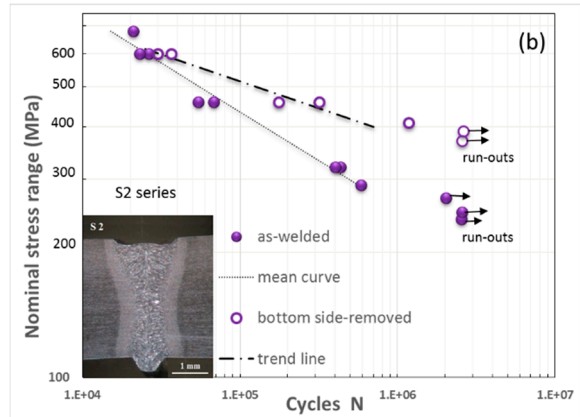

**Figure 7.** S-N curves for S1 (**a**) and S2 series (**b**) in two conditions: as-welded and bottom side-removed.

**Table 7.** Correlation between ratios of SCFs and fatigue strengths.

| Series | Fatigue Limits (MPa) (Imperfection) | | SCFs (Imperfection) | | Ratios | | |
|---|---|---|---|---|---|---|---|
| | as-Welded | Bottom Side-Removed | as-Welded | Bottom Side-Removed | Strength | SCF | Strengh/SCF |
| S1 | 180 (undercut) | 380 (underfill) | 3.29 (undercut) | 1.75 (underfill) | 2.11 | 1.88 | 1.12 |
| S2 | 270 (excess weld) | 390 (underfill) | 2.25 (excess weld) | 1.77 (underfill) | 1.44 | 1.27 | 1.13 |

For a better understanding of the behavior at relatively high stress levels, in addition to the as-welded condition, several specimens with bottom side-removed and two side-removed conditions were tested at 600 MPa. The fatigue lives of almost all samples at that stress level are shown in Figure 8. The results reaffirm, as expected, the increase in fatigue life due to the elimination of imperfections of weld beads. However, the lowest life of the specimens of the S2 series in the as-welded condition was due to the multiple small imperfections that practically formed a continuous crack along the width; for the group of specimens of all series in as-welded and bottom side removed conditions with similar fatigue lives in the range of 29,000 to 35,000 cycles, semi-elliptical cracks grew and covered the entire width or border of the specimens; from the specimen with a fatigue life of 36,587 cycles, all specimens presented a single semi-elliptic crack, where the smallest starts had a longer life than the largest ones. The exceptions were the specimens with starts in pores; thus, the fatigue life of 56,343 cycles corresponded to pores in the lateral side, meanwhile the fatigue life of 207,400 cycles correspond to internal pores. These results show the important effect of the quantity, size, and position of the imperfections on the fatigue life at high stresses and therefore the greater influence of the coalescence and crack growth at high stress levels.

*3.5. Fatigue Limit Assessment*

3.5.1. Local Properties

According to the fracture analysis, the fatigue started at the weld root and undercuts were located at the boundary between the columnar grains of the FZ and the CG-HAZ, while those in the underfill were located on the columnar grains of the FZ but were close to the HAZ as shown in Figure 9 in the macrostructure of a specimen of the S1 welded series. The fatigue paths were through the FZ and HAZ. Since the series were welded

with different heat inputs, the actual microstructures, hardness, and residual stresses were important factors in the fatigue strength. Once the heat input range used in the present work is similar to the range considered in a previous work [31], softened ferritic-bainitic in FG-HAZ and bainitic-martensitic in CG-HAZ and in FZ are expected as microstructures.

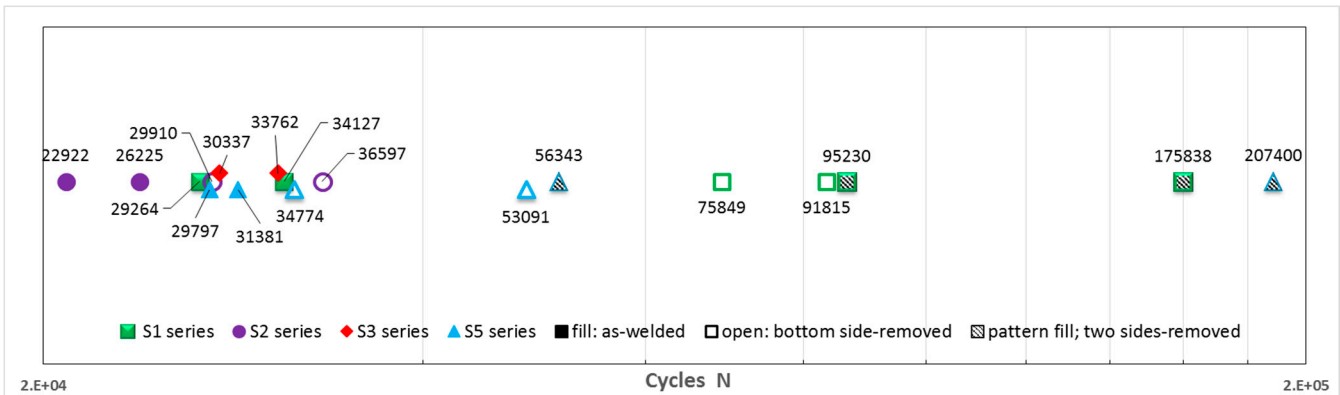

**Figure 8.** Fatigue lives of specimens at 600 MPa in various conditions.

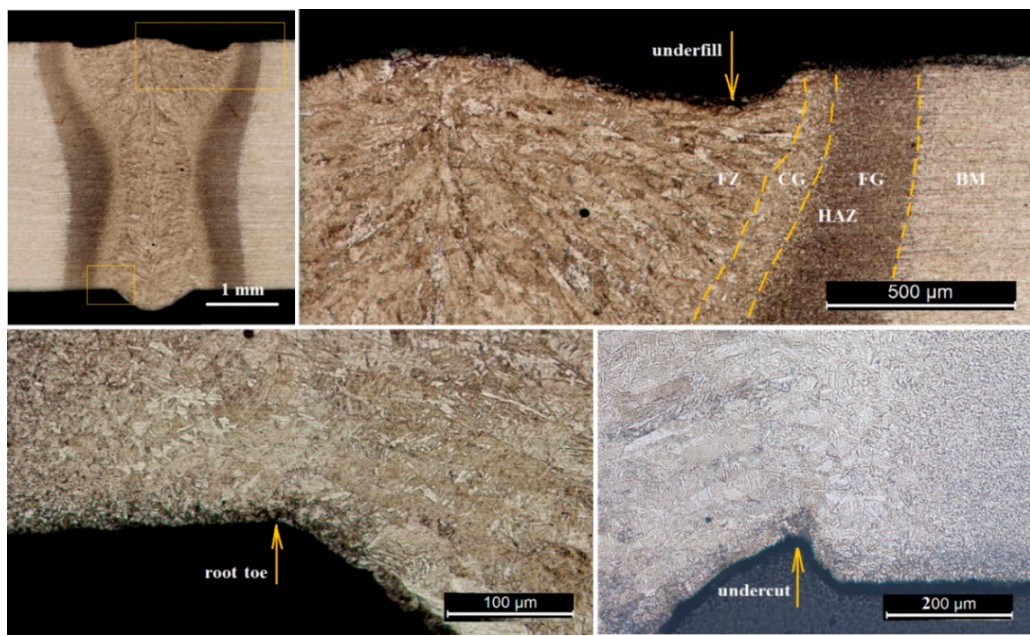

**Figure 9.** Fatigue initiation sites in the macrostructure for underfill, root toe, and undercut.

The microhardness (HV, 0.5 kg, 10 s) profiles were determined near to the surface on the cross-section of the weld beads for all welded series. Figure 10 shows two examples of the hardness profile found for S1 and S5 series. As can be seen from the hardness profile, in the FZ, the hardness is high and roughly the same for both sides of the weld bead at the fatigue sites, meanwhile in the HAZ, S1 series had softening, unlike S5 series, which practically did not have any. According to the results at the fatigue initiation sites, the hardness ranges for the welded series were 335 to 370 HV. Table 8 presents the local properties of hardness assumed at the sites where the cracks started and propagated for the fatigue strength assessment, which indicates minor differences between the welded series.

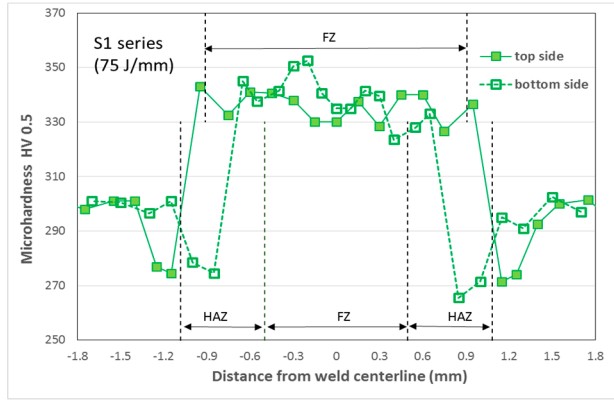 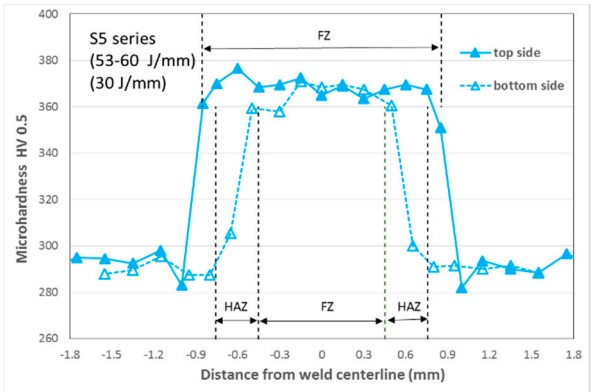

**Figure 10.** Microhardness profiles in S1 and S5 welded series.

**Table 8.** Data for fatigue strength assessment based on the modified Goodman line.

| Case | Condition | Series Imperfection | $K_t$ | $\varrho$ | $K_f$ | Hardness | Residual Stress | Fatigue Limits | $n_G$ Factor | |
|---|---|---|---|---|---|---|---|---|---|---|
| | | | | (mm) | | (HV) | (MPa) | (MPa) | RS [1] | no RS |
| 1 | as-welded | S1-undercut | 3.29 | 0.06 | 2.33 | 335 | 130 | 180 | 1.08 | 1.60 |
| 2 | bottom side-removed | S1-underfill | 1.75 | 0.53 | 1.60 | 335 | 130 | 380 | 0.90 | 1.10 |
| 3 | as-welded | S2-excess weld | 2.25 | 0.06 | 1.74 | 345 | 90 | 270 | 1.20 | 1.47 |
| 4 | bottom side-removed | S2-underfill | 1.77 | 0.60 | 1.63 | 345 | 90 | 390 | 0.94 | 1.09 |
| 5 | as-welded | S3-undercut | 4.41 | 0.03 | 2.74 | 350 | 50 | 210 | 1.05 | 1.22 |
| 6 | as-welded | S5-underfill | 1.95 | 0.27 | 1.74 | 370 | 200 | 215 | 1.17 | 1.90 |
| 7 | no-welded | Base metal | — | — | — | 295 | — | 600 | — | 0.98 |

[1] RS: residual stresses.

### 3.5.2. Residual Stresses

The residual stress field induced by the welding is one of the main factors affecting the fatigue strength. Figure 11 shows the results of residual stresses measured with X-ray diffraction technique in three fatigue specimens. Because initial measurements of the transverse and longitudinal residual stresses measured in one specimen (S1 series) showed similar distributions between transverse and longitudinal stresses, and to conservatively consider the effect of residual stresses in the two additional fatigue specimens (S3 and S5 series), only the longitudinal residual stresses were measured. As can be seen in Figure 11, there was an M-shape close to the HAZ and FZ with residual tensile stresses at the HAZ and tending to zero or compressive stresses at the weld centerline. These results, namely the shape, and the similar distribution between longitudinal and transverse residual stresses, agree with those reported in [35], in which the residual stresses of a laser-welded 4 mm high strength low alloy HSLA steel plate were measured and simulated by FEM. There, the residual stresses varied approximately from −200 MPa in the FZ to 250 MPa in the HAZ.

Focusing attention on the sites where the starts of fatigue occurred: in the inner border of the HAZ (for excess weld and undercuts) and in FZ, but near to HAZ (the lowest point of the underfill), residual stresses in the range of 200 to −40 MPa are observed in the graphs and although the differences in the values of the residual stresses between the welded series are small, they are approximately 200 to 80 MPa for S5 series, 130 to 50 MPa for S1 series, and 50 to −40 MPa for S3 series. The results suggest that as the heat input decreases and therefore the cooling rate increases, the residual tensile stresses progressively change to residual compressive stresses. These results may be due to the effect of strains by the bainite and martensite transformation [36], since the proportion of these microstructures increases as the heat input decreases. Table 8 shows the residual stresses assumed for each series.

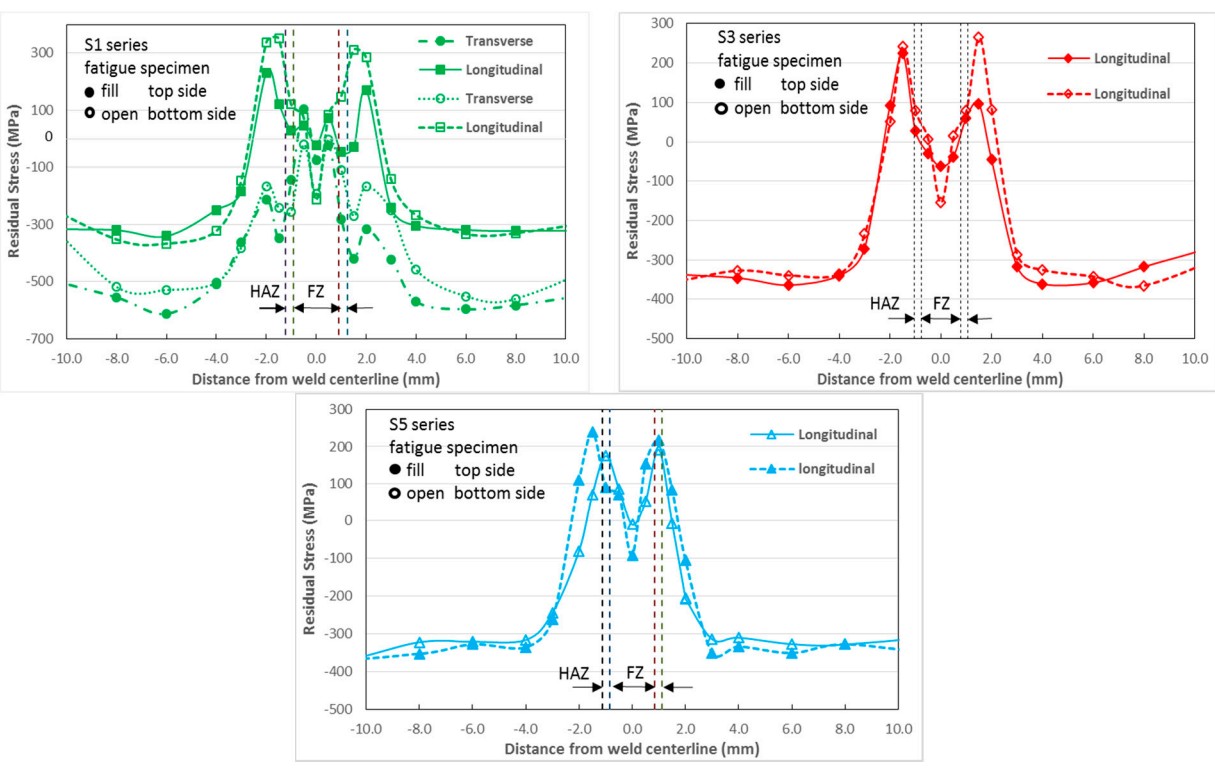

**Figure 11.** Residual stresses profiles in fatigue specimens.

### 3.5.3. Fatigue Limit Evaluation

In previous sections, it was shown that for low stress levels, the approach of reducing fatigue strength through stress-concentrating effect can be used. To consider the effect of the mean stress in the analysis of the fatigue strength at two million cycles of the welded series, due to its simplicity, the modified Goodman line will be used according to the expression:

$$\frac{K_f \, \sigma_a}{S_e} + \frac{K_f \, \sigma_m}{S_{ut}} = \frac{1}{n_G} \tag{2}$$

where $\sigma_a$ and $\sigma_m$ are the nominal alternating and mean stresses applied and both are equal to $\Delta\sigma/2$ (according to stress ratio R = 0); $K_f$, is the effective stress concentration factor calculated using the relationship of sensitivity to notch proposed by Peterson [37]; $S_{ut}$ and $S_e$ are the ultimate strength and the fatigue limit of the materials that will be estimated for each welded series and BM through the hardness HV, as 3.0 × HV and 1.5 × HV, respectively, and $n_G$ is the factor that evaluates the proximity to the mentioned failure criteria of the local stresses at imperfections of the specimens. When the residual stresses are considered, they will be added to the mean applied stress $\sigma_m$.

Table 8 shows the data used for the determination of the $n_G$ factors for both (with residual stresses and without residual stresses for the welded series) conditions and imperfections indicated there. Figure 12 illustrates the position of the fatigue failure points of the specimens according to the mean and alternating local stresses, in relation to respective modified Goodman lines. When examining the concordance between the assumed failure criterion and the cases considered, it was found that it is excellent in the case of the BM, generally good for all cases of the welded series when residual stresses were included and underestimated when they were not included; see Table 8 and Figure 12. There is a good agreement with and without residual stress when the $K_f$ values are low for underfill (cases 2 and 4), and in contrast to the highest values of $K_f$ (cases 1 and 5). Particularly for the excess weld and the underfill of the S2 and S5 series, respectively, there is the greatest discrepancy (close to 20%) between the experimental results and the assumed criterion (cases 3 and 6) when the residual stresses are considered.

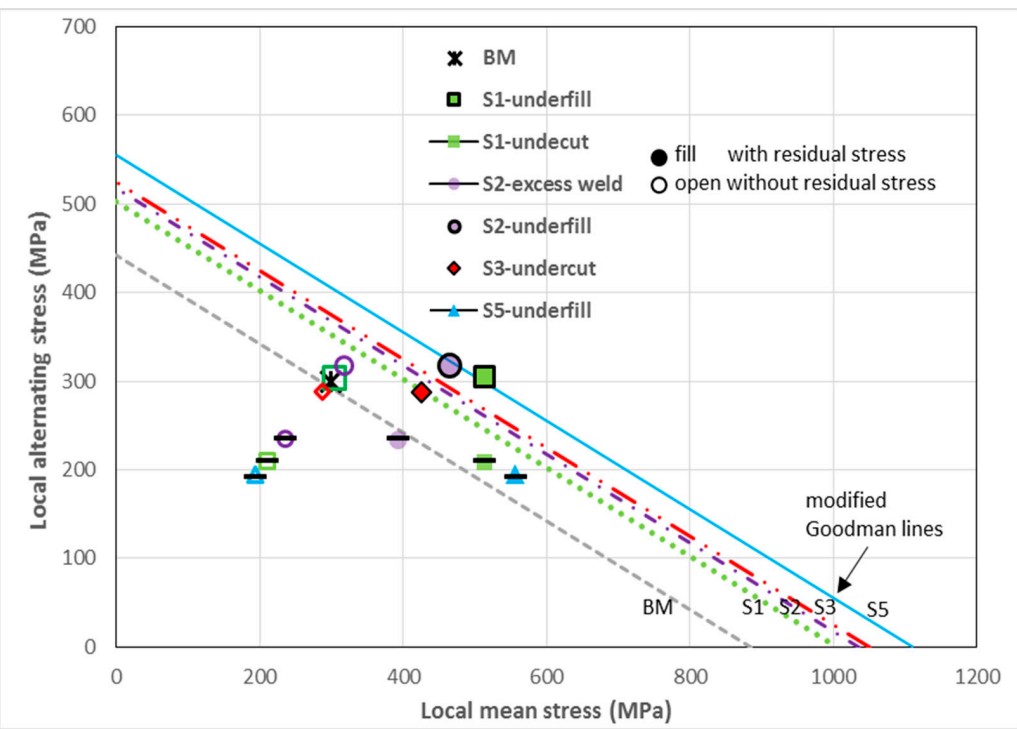

**Figure 12.** Fatigue failure points of specimens and modified Goodman lines.

The effects of hardness, residual stresses, and stress concentration on each welded series in the as-welded condition can be described as follows: the S1 series presented the lowest fatigue strength due to the combination of a high SCF at the undercut, a medium value of residual stress, and a relatively low hardness. The S5 series, despite having the highest hardness and although the SCF was medium, due to the high value of the residual stresses, it also presented low fatigue strength. The S3 series, with a low value of residual stresses and a high hardness, did not achieve a greater strength to fatigue because it had the highest SCF due to the sharp undercut in the weld root. The S2 series achieved the highest fatigue strength of all series with medium SCF and hardness values and a relatively low residual stress value. It can be inferred that this series can achieve greater fatigue strength if the multiple small imperfections present along the weld root decrease. In other works, the adverse effect of many small imperfections in addition to the weld bead profile on fatigue strength has also been reported [38,39].

Regarding the fatigue limits achieved by S1 (380 MPa) and S2 series (390 MPa) in the bottom side-removed condition, although the fatigue limit of the S2 series is expected to be higher because its hardness is greater and the residual stresses are lower, the slight difference may be due, as already pointed out in a previous section, to the greater depth of the underfill and to the effect combined with the weld ripples. In the case of the behavior of the S5-F9 specimens, which showed considerably greater fatigue strength than the other specimens of the S5 series, the weld profile in the bottom side was examined with more detail. It was found that the underfill values were below the average of the series and that the variations along the weld axis were small, as illustrated in other work [28]. The latter implies that the SCF of this series is lower and therefore its fatigue strength is higher.

The previous analysis showed that the SCF is a good predictor of fatigue limit in absence of multiple small imperfections, such as those of S2 series that were not easily detected or when the SCF is properly calculated (which was not the case of the S5 series). Conversely, it shows a great effect of the residual stresses that is magnified by the SCF and by the mean applied stress. In summary, although the residual stresses found can be considered small, they adversely affect the fatigue strength of the welded specimens, as well as the high SCFs due to the weld bead geometry and the small associated imperfections.

Meanwhile, the high hardness in the FZ and CG-HAZ allows the improvement of the strength of the welded series.

### 3.6. Fatigue Limit Predictions

The estimates at low stresses showed that the fatigue strength can be predicted if the hardness, residual stresses, weld geometry, and imperfections of the welded joints are known. Therefore, using the expression (2) with $n_G$ = 1 and the data in Table 8, the fatigue strength at two million cycles in terms of stress range was predicted and the corresponding results for each welded series and conditions are shown in Table 9 together with the experimental results and the relation that compares both results. Considering all the results, there is a good agreement between the experimental and predicted values. However, a difference of greater than 15 percent was found for the S2 and S5 series. For the S2 series, this can be attributed, as already noted, to imperfections in the weld root that the stress concentration factor does not consider. For the S5 series, the difference may be the result of a welding speed range (1.75 to 2.00 m/min) and therefore the value of the SCF, which can actually be higher than the one used ($K_f$, 1.95) since the latter was determined based on the average dimensions of the weld beads and at small imperfections in the weld ripples.

**Table 9.** Fatigue limit predictions.

| Series-Imperfection | Condition | Fatigue Limits | | |
|---|---|---|---|---|
| | | Experimental | Predicted | Ratio |
| S1-undercut | As-welded | 180 | 201 | 0.90 |
| S1-underfill | Bottom side-removed | 380 | 332 | 1.14 |
| S2-excess weld | As-welded | 270 | 336 | 0.80 |
| S2-underfill | Bottom side-removed | 390 | 363 | 1.07 |
| S3-undercut | As-welded | 210 | 222 | 0.95 |
| S5-underfill | As-welded | 215 | 292 | 0.74 |

Additionally, regarding the above predictions, and to explain the effect of the imperfections present in the weld root of the S2 series and due to the small size of the undercuts present in the S1 series, the Murakami's relations proposed to predict the fatigue limit were applied to the mentioned series. The relations have the form:

$$\sigma_w = \frac{1.43\,(HV + 120)}{\sqrt{Area}^{1/6}} \left[ \frac{1-R}{2} \right]^{\propto} \tag{3}$$

$$\propto = 0.226 + HV/10{,}000, \tag{4}$$

where $\sigma_w$ (in MPa) is the fatigue limit, $HV$ is the Vickers hardness, $R$ is the stress ratio and $\sqrt{Area}$ (in μm) is the size parameter of the imperfection. The data shown in Table 10 corresponds in the case of the S1 series to the average size of undercuts, while in the case of the S2 series, it corresponds to the average depth of 10 measurements on the long and narrow imperfections, which are practically continuous in the weld root. In both cases, the $\sqrt{Area}$ parameter was calculated as $\sqrt{Area} = a\sqrt{10}$, where $a$ is the crack depth, according to Murakami's recommendation for this type of imperfection (shallow surface crack) [12]. In the stress ratio, the residual stresses of Table 8 were considered and through an iterative process, in which fatigue limit values were assumed until equality in expression (3) was reached, the fatigue limits were predicted, which are shown in Table 10 as well as the respective comparisons with the experimental results.

**Table 10.** Data and parameters for prediction of fatigue limits.

| Series Imperfection | Crack Sizes Length, Depth | $\sqrt{Area}$ | Fatigue Limits | | |
|---|---|---|---|---|---|
| | (mm, μm) | (μm) | Experimental (MPa) | Predicted (MPa) | Ratio |
| S1-undercut | 1.11, 40 | 126 | 180 | 214 | 0.84 |
| S2-shallow crack | >0.23, 22 | 70 | 270 | 254 | 1.06 |

The results of Table 10 show that adequate predictions can be achieved through the approach proposed by Murakami. In the case of the S2 series, the prediction is better adjusted to the experimental value, while in the case of the S1 series, a similar result was obtained. However, in the predictions based on this last approach, it is necessary to know the imperfection and its size, which may not be evident, as in the case of the S2 series, where only after the fracture of the specimens, they can be observed. Conversely, there may be variations in the sizes of the imperfections, which requires measurements and the application of some criterion that in the present case, was the average size. In the case of undercuts (for series S1), a better prediction can be achieved considering that 4 to 5 undercuts can be present in each specimen; therefore, it is more likely that the size is greater than the average size (40 μm). Considering that the depth of the undercuts varied from 15 to 105 μm [28], for example, an undercut depth of 90 μm, the fatigue limit will be 180 MPa, which is the same as the experimental result.

## 4. Conclusions

For four laser butt joints in a thin HSLA steel plate, the effects of the weld profile, imperfections, hardness and residual stresses were considered to explain the experimental S-N curves and for predictions at low stress levels, applying approaches based on both stress-concentrating effect and Murakami's relationship. The main conclusions drawn are:

- The fatigue strength of the welded series exceeded the FAT100 reference curve with fatigue limits in the range of 180 to 340 MPa, presenting multiple imperfections as shallow but sharp, and considered as crack-like imperfections and with B and D quality levels according to the ISO13919-1 standard;
- Although each series presented several fatigue failure modes due to the presence of different imperfections in the weld bead, a dominant failure mode strongly related to the imperfection with the highest SCF was observed for each series: 3.29—undercut—S1, 2.25—excess weld—S2; 4.41—undercut—S3; 1.95—underfill—S5;
- At low stress levels, the local properties of each welded series: hardness, residual stress, and weld bead geometry, determined the fatigue limits and their trends were in good agreement with the experimental results. Both high SCF and relatively low residual tensile stresses (<200 MPa) affected the fatigue limits; only when SCFs are low (<2.0) can the level (<130 MPa) of residual tensile stresses not be considered, while the local hardness increased the fatigue limits of the welded series although its influence is lower than that of the other two factors;
- The predictions of the fatigue limits were acceptable by the stress-concentrating effect when there were no small imperfections associated with excess weld, undercut, or underfill, while when these were present or for small undercuts, the relationship proposed by Murakami was appropriate.

**Author Contributions:** Conceptualization, P.G.R. and J.F.; methodology, P.G.R., A.C.B., C.C., and J.F.; software, P.G.R.; validation, P.G.R., F.A., and A.C.B.; formal analysis, P.G.R.; investigation, P.G.R.; writing—original draft preparation, P.G.R.; writing—review and editing, F.A., A.C.B., and C.C.; supervision, J.F. All authors have read and agreed to the published version of the manuscript.

**Funding:** The authors acknowledge the support provided by the Universidad de las Fuerzas Armadas-ESPE for the scholarship and Mário da Costa Martins and Fernando Meireles of the Orthopedia Medica Company for laser welding manufacturing. This research is sponsored by FEDER funds through the program COMPETE–Programa Operacional Factores de Competitividade— and by national funds through FCT–Fundação para a Ciência e a Tecnologia–, under the project UIDB/00285/2020.

**Institutional Review Board Statement:** Not applicable.

**Informed Consent Statement:** Not applicable.

**Data Availability Statement:** Not applicable.

**Conflicts of Interest:** The authors declare no conflict of interest.

## Nomenclature

| | |
|---|---|
| $a$ | Crack depth |
| $\sqrt{Area}$ | Murakami's area parameter |
| $C$ | parameter of the S-N curve |
| $HV$ | Vickers hardness |
| $m$ | Slope of S-N curve in log-log scales |
| $N$ | Number of cycles |
| $K_t$ | Stress concentration factor |
| $K_f$ | Effective stress concentration factor |
| $R$ | Stress ratio equal to minimum stress divided by maximum stress |
| $n_G$ | Goodman criteria factor |
| $S_{ut}$ | Tensile ultimate strength |
| $S_e$ | Fatigue limit |
| $\Delta\sigma$ | Nominal stress range |
| $\varrho$ | Imperfection radius |
| $\sigma_a$ | Nominal alternating stress |
| $\sigma_m$ | Nominal mean stress |
| $\sigma_w$ | Fatigue limit according Murakami's expression |
| $\propto$ | Murakami's factor |

**Abbreviations**

| | |
|---|---|
| ASTM | American Society for Testing and Materials |
| BM | Base metal |
| CG-HAZ | Corse-grain heat-affected zone |
| FAT | Fatigue class curve |
| FEM | Finite element method |
| FG-HAZ | Fine-grain heat-affected zone |
| FCG | Fatigue crack growth |
| FCGR | Fatigue crack growth rate |
| FM | Failure mode |
| FZ | Fusion zone |
| HAZ | Heat-affected zone |
| HSLA | High-strength low-alloy |
| HV | Hardness Vickers |
| IIW | International Institute of Welding |
| ISO | International Organization for Standardization |
| NDT | Nondestructive testing |
| HI | Heat input |
| SCF | Stress concentration factor |

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
