# Peer review of "Fatigue Performance of Thin Laser Butt Welds in HSLA Steel"

_metals, doi:10.3390/met11101499_

Round 1

Reviewer 1 Report

This paper studies the effect of various factors on the fatigue strength of laser-welded butt joints in thin high-strength low-alloy (HSLA ) steel. The effects of the weld profile, imperfections, hardness and residual stresses were considered to explain the results found in the S-N curves of four welded series. The analysis of fatigue behavior at low stress levels through the stress-concentrating effect and by the relationship proposed by Murakami showed good agreement with the experimental results. The results showed acceptable fatigue strength even though the weld series presented multiple-imperfections.
The paper is well-written, but I have the following concerns:  
• In the case of data from tables 1 and 2, I suggest not to refer to the publication, which is not in the open acces but to provide brief information about the tests performed.
•    The authors should also provide more information about the failure modes of the different weld series. 
•    The authors should provide more detailed analysis of the results . In particular , it is not clear what is the difference between the S2 and S3 curves
•    The author should provide the results on the confidence F-test of the results obtained. 
In conclusion, I believe that the reviewed material presents very interesting and applicable information on the fatigue properties of welds in HSLA steels.

Reviewer 2 Report

This paper is focused on understanding the significant factors affecting the fatigue strength of laser-welded butt joints in thin high-strength low-alloy (HSLA) steel. The effects of the weld profile, imperfections, hardness and residual stresses were considered to explain the results found in the S-N curves of four welded series. The literature review in the Introduction is very thorough. However, the authors could give some examples of the practical applications of laser-welded thin HSLA steel. The experimental work would appear to have been competently executed. The fatigue limits of the welded series predicted through the stress-concentrating effect and through the relationship proposed by Murakami showed reasonably good agreement with the experimental results. The English is poor and needs a great deal of polishing. I have provided some of the most obvious corrections below:-

Page 1, Abstract, Line 13: …factors on the… -> …factors affecting the…

A Nomenclature at the beginning would be helpful.

In particular, define stress ratio R as minimum stress divided by maximum stress in a cycle.

Page 1, Line 25: The high-strength… -> High-strength…

Page 1, Line 40: Missing text.

Page 1, Line 43: …in function… -> …as a function…

Page 1, Line 43: …defects and… -> …defects, and…

Page 1, Line 46/Page 2, Line 47: , however they… -> ; however, they…

Page 2, Line 56: …of the material, -> …of the material;

Page 2, Line 59: …features as… -> …features such as…

Page 3, Line 104: …is close the… -> …is close to the…

Page 3, Table 1: Add (wt%) to caption.

Page 3, Table 2: Caption missing.

Page 4, Line 115: Square brackets missing from reference number.

Page 3, Line 121: …was suplied only… -> …was supplied only…

Page 4, Line 129: …be consider as… -> …be considered as…

Page 4, Line 141: …carefully grinded… -> …carefully ground…

Page 4, Line 151: …levels, however, when… -> …levels; however, when…

Page 5, Line 186: …stress ranged… -> …stress range…

Page 5, Line 197: …The FAT100 curve is clearly above… Shouldn’t this be below?

Page 5, Line 202: …similar than that… -> …similar to that..

Page 5, Line 205: …The FAT100 curve is clearly above… Again, shouldn’t this be below?

Page 7, Line 241: …series, there… -> …series; there…

Page 8, Table 5: This is very confusing. It is unclear which legends apply to which optical micrograph. Also, the higher magnification shots have no distance markers.

Page 9, Line 359: …the failures modes… -> …the failure modes…

Page 10, Line 400: …behaviour; the… -> …behaviour, the…

Page 13, Line 488: …side in the… -> …side on the…

Page 14, Line 542: …beads. however, -> …beads. However,

Page 14, Line 551: …side meanwhile… -> …side, meanwhile…

Page 15, Figure 9: Which specimen(s) are being referred to in this figure?

Page 16, Line 645: …transversal… -> …transverse…

Page 16, Line 659: …that are… -> …that these are…

Page 17, Figure 11: On the first graph, Transversal -> Transverse

Page 18, Line 726: …even though when… -> …even when…

Page 19, Line 760: …along to the axis weld… -> …along the weld axis…

Page 19, Line 765: …SCF it is… -> …SCF is…

Page 20, Line 801: …hardness Vickers… -> …Vickers hardness…

Page 21, Line 850: …associated to excess… -> …associated with excess…

Page 22, Line 927: Reaproved -> Reapproved

Reviewer 3 Report

Review of paper “Fatigue performance of thin laser butt welds in HSLA steel” by Riofrio et al.

General

The investigation has manufactures and fatigue tested samples of 3 mm HSLA steel sheet containing a transverse laser butt weld.  S-N curves  of a number of different weld procedures were compared together with fatigue data on the original parent sheet. Detailed studies were made of weld microhardness, residual stress fields and local macrostructures at the sites of fatigue crack initiation. Defects at the sites of fatigue crack initiation were characterised and local stress concentrations calculated. A model to predict the fatigue limit stress amplitude was developed based on local crack initiation approaches using as  inputs defect Kt values, residual stresses, hardness measurements, using the Goodman rule to apply mean stress corrections.  

The paper contains interesting results and the investigation is detailed and rigorous. It is in principle worthy of publication, but there are a number of issues requiring clarification or amendment, before it can be published.

Although the paper itself is well produced, the quality of English is poor, and in places obscures the meaning of the text. Some of the errors are picked out below, but it would be best if the entire paper was edited to correct the English.

Specific points

  • P1 L25 Insert “to crack initiation” after “fatigue resistance”
  • P1 L39 There is a sentence missing here? Please insert.
  • P1 L43 Please correct English. Should read “or cracks as a function of the hardness…..”
  • P1 L44 Correct English “…the effect of the small defects…”
  • P1 & 2 Lines 40-81 paragraph is too long. Break it up.
  • P2 L56 Provide cited references for statement that that fcgr is insensitive to mechanical strength. Put full stop after “material”.
  • P2 L57 Start new sentence “However….”
  • P2 L63 Begin new paragraph “Thus….”
  • P2 L64 Delete “Meanwhile”. Begin sentence “In [23] it is shown…”
  • P2 L67 Insert full stop after [5]. Begin new sentence ”Fatigue..”
  • P2 L68 full stop after “exceeded”. Delete “however”, begin new sentence  “Although…”
  • P2 L70 Begin new paragraph “The general effect….”.
  • P2 L82 Delete “As reviewed”. Begin sentence “There are ….”
  • P2 L84 Delete “Therefore” Begin sentence “In this study….”. Rewrite this sentence- it is far too long and not comprehensible as it is.
  • P3 Table 1 Insert “wt% into the title
  • Table 2 Insert correct title;
  • Table 2 Did this material exhibit an identifiable yield point or are these values proof strength properties? If proof strength insert criterion (0.1%, 0.2% etc.)
  • Table 2 Please justify the precision implied by 5 significant figures and 0.01 of a MPa unit for the strength values, and 4 significant figures for %elongation. If it cannot be justified, amend the data to a realistic number of significant figures.
  • P3 Insert [] around reference 31.
  • P4 L127-129 rephrase meaning very unclear.
  • P4 L141 delete grinded, Insert “ground”
  • Figure 1 a & b state units for dimensions in caption.
  • P5 section 2 Measurement of residual stress. Indicate the position, extent. and direction of the residual stress measurement traverse. perhaps on figure 1 b or a separate figure. State if just a single traverse was made or a number of them. State the spacing of the measurements, and the dimensions of the area over which the residual stress field will be averaged.
  • P5 L186 “….The nominal stress range….”
  • P5 L187 “…which corresponds to “
  • Table 4 Can the precision of the values of log C and m (5 and 4 significant figures) be justified in the light of the scatter in the fatigue data points.? Please consider and amend. Amend heading on fatigue limits column to read “Stress range at fatigue limit (MPa).
  • P 8 & 9 Table 5- are the pictures part of table 4 or are they part of figure 4? This is very confusing. Please amend to clarify.
  • P10 Section 3.3 L401. Correct “Finite elements” to Finite element.
  • P10 Description of FE process. Provide brief detail of how the dimensions of the defects were measured, and the extent of dimension variation from sample to sample. Was a sensitivity study made to explore the effects on Kt of the measured variations? Please state in text whether this was done and the results if available.
  • FE model. please state whether model was 2 D or 3D and whether plane stress or plane strain and describe mesh sensitivity studies performed to check if mesh was correctly dimensioned. 
  • P11 Table 6 Please define the reference stress used in calculating Kt.
  • P12 L445/446 Figure 6 state the applied load used in the FE calculations.
  • P14 Table 7 explicitly define the strength and SCF ratios.
  • P15 figure 9 Change the caption these are pictures not fatigue lives as stated.
  • P17 Figure 11. Please insert lines corresponding to zero stress and zero distance on all 3 figures to improve clarity.
  • P17 In describing the residual stress results please comment on the level of biaxiality in the residual stress field, and the size of the transverse field in comparison with the longitudinal one, and between the topside and bottom side. Comment on the errors in stress measurement arising from averaging in the measurement area in regions of rapidly spatially changing stress field such as the region of the initiation sites. Comment on how this will influence the accuracy of the model for fatigue threshold calculation. Please indicate on figure 11 the location of initiation points as shown in figure 9.
  • P17 clarify how Se and Sut were calculated.
  • P17 Justify use of Goodman mean stress correction which is well known for being conservative, instead of the more accurate Gerber parabola.
  • P17 Clarify which strength values were used for calculation of Kf values from Kt. These should be displayed in table 8.
  • Clarify the calculation and use of the parameter nG as well as the procedure for calculation of ultimate strength SUT and fatigue limit Se from hardness values. Strength and fatigue limit will be dependent on microstructure as well as hardness, and no information is provided about these. Please expand.
